# Tomato Fruit Nutritional Quality Is Altered by the Foliar Application of Various Metal Oxide Nanomaterials

**DOI:** 10.3390/nano12142349

**Published:** 2022-07-09

**Authors:** Jesus M. Cantu, Yuqing Ye, Jose A. Hernandez-Viezcas, Nubia Zuverza-Mena, Jason C. White, Jorge L. Gardea-Torresdey

**Affiliations:** 1Department of Chemistry and Biochemistry, The University of Texas at El Paso, 500 West University Avenue, El Paso, TX 79968, USA; jcantu3@miners.utep.edu (J.M.C.); yye@miners.utep.edu (Y.Y.); jahernandez19@utep.edu (J.A.H.-V.); 2Environmental Science and Engineering Ph.D. Program, The University of Texas at El Paso, 500 West University Avenue, El Paso, TX 79968, USA; 3Connecticut Agricultural Experiment Station, New Haven, CT 06511, USA; nubia.zuverza@ct.gov (N.Z.-M.); jason.white@ct.gov (J.C.W.)

**Keywords:** ferrite hybrids, ZnO, Mn_3_O_4_, tomato, carbohydrates, phytonutrients

## Abstract

Carbohydrates and phytonutrients play important roles in tomato fruit’s nutritional quality. In the current study, Fe_3_O_4_, MnFe_2_O_4_, ZnFe_2_O_4_, Zn_0.5_Mn_0.5_Fe_2_O_4_, Mn_3_O_4_, and ZnO nanomaterials (NMs) were synthesized, characterized, and applied at 250 mg/L to tomato plants via foliar application to investigate their effects on the nutritional quality of tomato fruits. The plant growth cycle was conducted for a total of 135 days in a greenhouse and the tomato fruits were harvested as they ripened. The lycopene content was initially reduced at 0 stored days by MnFe_2_O_4_, ZnFe_2_O_4_, and Zn_0.5_Mn_0.5_Fe_2_O_4_; however, after a 15-day storage, there was no statistical difference between the treatments and the control. Moreover, the β-carotene content was also reduced by Zn_0.5_Mn_0.5_Fe_2_O_4_, Mn_3_O_4_, and ZnO. The effects of the Mn_3_O_4_ and ZnO carried over and inhibited the β-carotene after the fruit was stored. However, the total phenolic compounds were increased by ZnFe_2_O_4_, Zn_0.5_Mn_0.5_Fe_2_O_4_, and ZnO after 15 days of storage. Additionally, the sugar content in the fruit was enhanced by 118% and 111% when plants were exposed to Mn_3_O_4_ and ZnO, respectively. This study demonstrates both beneficial and detrimental effects of various NMs on tomato fruit quality and highlights the need for caution in such nanoscale applications during crop growth.

## 1. Introduction

With the global population increasing, agricultural production will need to expand by over 60% by the year 2050 in order to sustain global food security [1]. To date, conventional agrochemicals have been widely used in efforts to achieve food security. However, these currently used practices suffer from highly inefficient delivery and utilization, resulting in wasted energy and water, as well as a compromised output [2]. The need to increase the efficiency of food production has led to a great interest in the use of nanotechnology to improve crop yield by using nanofertilizers as a strategy for precision agriculture. Nanomaterials have the potential to enhance agricultural production by promoting plant nutrition compared to traditional fertilizers [3,4].

A robust literature has developed that is assessing the potential applications and implications of various metal-based nanomaterials. For instance, in a study with seedlings, Ye et al. reported that Mn-nanopriming (nanoseed treatment prior to germination) improved root growth by approximately 33% and 55% in both pristine water and brine water, respectively [5]. Moreover, the application of μCuO at 20 mg/kg to soil showed an activation of both chlorophyll a and catalase in sugarcane; all treatments (kocide^®^, Cu nanoparticles, μCuO, and CuCl_2_) caused Cu accumulation (47–269%) in sugarcane root tissues at all concentrations tested (20, 40, and 60 mg/kg) [6]. Imperiale et al. found that the introduction of CdS and ZnS quantum dots at doses of 10–30 mg in hyperaccumulating plants *N. cearulescens* and *A. halleri* caused an increase in fresh weight and an accumulation of Zn and Cd in the aerial tissues [7].

Tomatoes (*Solanum lycopersicum* L.) are important global food, with production exceeding over 40 million tons in 2020 [8]. The fruits are processed into various products, such as tomato sauce, juice, and paste [9]. Moreover, tomatoes are rich in carbohydrates, phytonutrients, vitamins, and micronutrients. Previous studies have shown variable results from metal-oxide nanomaterial application on tomato plant growth and yield. Raliya et al. applied TiO_2_ and ZnO to tomato plants through both foliar and soil exposure and showed an increase not only in fruit production but also in lycopene by 113% and 80% at doses of 100 mg/kg of ZnO and TiO_2_, respectively [10]. Barrios et al. and Adisa et al. reported similar increases in tomato fruit yield upon exposure to pristine CeO_2_ via both soil and foliar applications [11,12]. However, others have reported decreases in tomato fruit yield upon exposure to nanoscale CeO_2_ at 1 and 10 ppm [13]. Akanbi-Gada et al. observed an antagonistic effect on bioactive compounds (total phenols, flavonoids, β-carotene, and lycopene) of tomato fruits when plants were treated with ZnO at various concentrations (300–1000 mg/kg) by soil application [9]. Yue et al. recently reported that a foliar application of MnFe_2_O_4_ to tomato plants promoted early flowering and increased the overall fruit yield as well as fruit size and weight [8]. Additionally, MnFe_2_O_4_ exposure altered the metabolite profile, including increased levels of rutin, quercetin, glucose-6-phosphate, salicylic acid, and phenylalanine, thereby improving fruit quality [8].

Currently, there is little work investigating the effects of hybrid ferrite materials on tomato plant growth and fruit quality. Here, hybrid ferrite materials were synthesized via a coprecipitation/dehydration method and characterized via X-ray diffraction (XRD), scanning electron microscopy (SEM), transmission electron microscopy (TEM) and dynamic light scattering (DLS). Tomato (*S. lycopersicum*) Candyland red variety was chosen as a model plant to assess the effects of hybrid ferrites (Fe_3_O_4_, MnFe_2_O_4_, ZnFe_2_O_4_, and Zn_0.5_Mn_0.5_Fe_2_O_4_), hausmannite (Mn_3_O_4_), and wurtzite (ZnO) materials on the plant growth and fruit quality. Tomatoes were exposed to the various materials and ionic counterparts via foliar application and were grown to full maturity. The chlorophyll content was measured one week after each application and during the harvest while the plant growth, micronutrient uptake, fruit production, and phytonutrients were measured postharvest. This work increases our understanding of the safe and sustainable use on nanoscale metal oxides in agricultural production.

## 2. Methodology

### 2.1. Synthesis of Nanomaterials

The synthesis for the iron- and manganese-oxide nanomaterials was conducted following Arteaga-Cardona et al. [14]. We used a 1 L solution containing a total metal (M^+^) concentration of 30 mM as titrated at a flow rate of 0.1 mL/min with 90 mL of 1 M NaOH to produce a M^+^ to OH^-^ ratio of 1:3. For the zinc oxide, a M^+^ to OH^−^ ratio of 1:2 was used to form the nanomaterial as described by Flores et al. [15]. Once the titration was complete, the solution was heated to boiling for 1 h to produce the oxide form of the nanomaterial. The product was then centrifuged at 3500 RPM for 5 min at room temperature, washed, and rinsed using ultrapure water (UPW) with a resistance of 18.2 MΩ. The washing process was repeated 3 times to remove any impurities and byproducts formed during the reaction. After washing, the products were oven-dried at 60 °C overnight. The precursor and concentration ratios for each nanomaterial are shown in Table 1.

### 2.2. Nanomaterial Characterization

All nanomaterials (Fe_3_O_4_, MnFe_2_O_4_, ZnFe_2_O_4_, Zn_0.5_Mn_0.5_Fe_2_O_4_, Mn_3_O_4_, and ZnO) were analyzed using a Panalytical Empyrean diffractometer according to Cantu et al. [16]. The diffraction patterns were fitted using literature crystallographic data, FullProf suite software, and a Le Bail fitting [15,17,18,19,20].

The nanomaterials were imaged by scanning electron microscopy (SEM) through secondary electron mode using a Hitachi S-4800 according to Ye et al. [5]. The Fe_3_O_4_, MnFe_2_O_4_, ZnO, and Mn_3_O_4_ NMs were sputter-coated with gold using an SPI Module with a 18 mA current for 40 s. Moreover, the NMs were also imaged by transmission electron microscopy (TEM) using a Hitachi HT7800 with a LaB6 filament, high-resolution mode, an accelerating voltage of 80,000 kV, an emission 5–10 μA, and a vacuum of 6.8 × 10^−5^ Pa.

The hydrodynamic size and zeta (ζ)-potential for all nanomaterials were measured using a Malvern Panalytical Zetasizer Nano ZS90. In short, 250 mg/L NM suspensions were prepared in UPW and sonicated (Crest Ultrasonics 275DA) for approximately 20 min at room temperature. Thereafter, the hydrodynamic size and ζ-potential were measured three times for each material.

### 2.3. Tomato Plant Cultivation

Candyland red tomato seeds purchased from Harris seeds were sowed in vermiculite. Three days after sowing, the seeds started to germinate. The seedlings were allowed to grow in the vermiculite for 14 days prior to transplanting. On day 14, the seedlings were transplanted into pots filled with MiracleGro^®^ potting mix. Each pot contained four seedlings. After 35 days, the pots were thinned to one plant that was grown to full maturity. Nanomaterial suspensions and ionic (Zn, Fe, or Mn) counterparts of 250 mg/L were prepared and applied via foliar application. The treatments were applied during the vegetative stage (day 43, 32 mL) and early flowering stage (day 78, 48 mL) for a total of 80 mL after transplantation. The applications were conducted while keeping the soil covered. Each treatment consisted of four replicates (pots). Plants were grown for a total of 135 days. At harvest, tomato fruits were sectioned into three parts: one was stored at −80 °C, another was stored at room temperature for 15 d prior to storage at −80 °C, and the last was oven-dried for elemental analysis and carbohydrate quantification. Moreover, plant tissues including roots, stems, and leaves were separated for biomass determination and stored for elemental analysis.

### 2.4. Leaf Chlorophyll Content

The chlorophyll content was measured one week after each treatment application and at harvest using a Minolta SPAD (single photon avalanche diode handheld device) (Minolta Camera, Japan) in which five leaves were randomly measured from each treatment and replicate and were then averaged.

### 2.5. Tissue Elemental Analysis

The separated plant tissues were oven-dried for 72 h at 60 °C prior to grinding and homogenizing. Once homogenized, 0.2 g of each replicate and treatment were weighed and digested in a digestion block (SCP Science, Baie-d’Urfé, QC, Canada) using 5 mL of plasma pure HNO_3_ for 45 min at 115 °C. Thereafter, 2 mL of 30% hydrogen peroxide were added and the samples were redigested for an additional 20 min. Following the digestion, the samples were diluted to 25 mL using UPW and were analyzed by inductively coupled plasma optical emission spectroscopy (ICP-OES) using a PerkinElmer Optima 4300 DV (PerkinElmer, Shelton, CT, USA). In order to validate the digestion and analytical methods, blanks and standard reference materials were used (NIST-SRM 1570a and 1547; spinach and peach leaves).

### 2.6. Carbohydrate Extraction and Quantification

The total sugar content in the tomato fruits was determined according to DuBois et al. as optimized for microplate analysis [21]. In short, 10 mL of 80% ethanol was added to 100 mg of dried tomato fruit. The samples were placed in a hot water bath at 80 °C for 30 min followed by centrifugation for 20 min at 4500 RPM. The extract was then transferred into clean 50 mL conical vials and the extraction was repeated two additional times. The extracts were combined, evaporated to 3 mL, and diluted to 25 mL with UPW. The sugar extracts were analyzed using a ThermoFisher Multiskan Skyhigh microplate reader at 490 nm using glucose as a standard.

The fruit starch content was determined according to the optimized method of Verma and Dubey [22]. In short, the sugar extraction residue was dried in the oven for 24 h at 70 °C followed by homogenization with 2 mL of water. The mixture was placed in a hot water bath at 90 °C for 15 min. Thereafter, the samples were allowed to cool to room temperature prior to the incorporation of 2 mL concentrated H_2_SO_4_. The samples were vortexed for 15 min following centrifugation at 4500 RPM for 20 min. The supernatant was transferred to a 50 mL conical vial. The extraction was repeated twice more using 50% H_2_SO_4_. The supernatants were combined and diluted to 50 mL using UPW. The starch content was quantified using a ThermoFisher Multiskan Skyhigh microplate reader and measuring the absorbance at 490 nm using potato starch to construct a calibration curve (R^2^ ≥ 0.98) [21].

### 2.7. Bioactive Compound Assays

Carotenoid analysis was conducted according to Nagata and Yamashita [23]. Pigment extraction was conducted by homogenizing 1 g of frozen tomato fruit with 10 mL of a 4:6 acetone/hexane solution using a Thermolyne Speci-Mix test tube rocker for 10 min. The supernatant was then analyzed on a microplate spectrophotometer (Multiskan SkyHigh, Thermo Fisher Scientific, Cleveland, OH, USA) and absorbance was measured at 543, 505, 645, and 663 nm. Moreover, purchased tomato fruits (Sunset Sweet Bites) were used as a positive control, and the carotenoid content was calculated using Equations (1) and (2).
(1)$$Lycopene\;\left(\mathrm{mg}/100\mathrm{mL}\right)=-0.0458A_{663}+0.204A_{645}+0.372A_{505}-0.0806A_{453}$$
(2)$$\beta \;Carotene\;\left(\mathrm{mg}/100\mathrm{mL}\right)=0.216A_{663}-1.22A_{645}-0.304A_{505}+0.452A_{453}$$

Frozen tomato fruits were lyophilized using a Labconco Freezone4.5 at −40 °C and 0.293 mbar. Afterwards, the freeze-dried samples were ground and used for total phenolic compounds, and flavonoids extractions. The total phenolics were extracted and quantified as described by Singleton et al. [24]. One hundred mg of lyophilized tissue was homogenized with 500 μL of 1:1 acetone/water solvent in a 2 mL Eppendorf tube for 20 min. The samples were then centrifuged at 12,000 RPM for 10 min and the supernatant was recovered for analysis. For total phenolic compounds, 8 μL of Folin reagent, 200 μL of UPW, 20 μL of Na_2_CO_3_, and 4 μL of extract were transferred to each well in a microplate and incubated at 45 °C for 30 min. Thereafter, the absorbance was measured via a microplate spectrophotometer (Multiskan SkyHigh, Thermo Fisher Scientific, Cleveland, OH, USA) at 750 nm. Gallic acid was used as a standard for the calibration curve.

For the flavonoid assay, the method of Dow was modified and adapted for a microplate analysis [25]. Fifty milligrams of lyophilized tissue was homogenized with 5 mL of methanol (HPCL grade) for 10 min using a Thermolyne Speci-Mix test tube rocker. The supernatant was then stored for analysis. In short, 150 μL of 2% AlCl_3_ methanolic solution and 150 μL of extract were transferred to a microplate and allowed to rest in the dark for 20 min prior to measurement. The absorbance was then measured at 415 nm using a microplate spectrophotometer (Multiskan SkyHigh, Thermo Fisher Scientific, Cleveland, OH, USA) with quercetin as a standard for the calibration curve.

### 2.8. Statistical Analysis

A one-way ANOVA and Tukey’s HSD with a test error of *p* ≤ 0.05 were used to determine statistical significance of parametric data. For all data that followed nonparametric trends, a Kruskal–Wallis/Dunn’s test was used. Origin (OriginPro 2021b) and Minitab19 were used for all statistical analyses.

## 3. Results and Discussion

### 3.1. Nanomaterial Characterization

The diffraction patterns for the iron oxide materials (Fe_3_O_4_, MnFe_2_O_4_, ZnFe_2_O_4_, and Zn_0.5_Mn_0.5_Fe_2_O_4_), Mn_3_O_4_, and ZnO are shown in Figure 1. The iron oxide materials were found to have characteristic peaks for magnetite diffracted at 30.16°, 35.52°, 37.16°, 43.17°, 47.27°, 53.56°, and 57.10° in 2θ. These peaks correspond to the (220), (311), (222), (400), (331), (422), and (511) planes. Moreover, the ZnO was found to be in the wurtzite phase with characteristic peaks at 31.82° (100), 34.40° (002), and 36.32° (101) in 2θ (the planes are given in parenthesis). The Mn_3_O_4_ nanomaterial was found to be in the hausmanite phase with the characteristic peaks occurring at 28.91°, 30.98°, 32.38°, 36.07°, 38.09°, 44.39°, and 50.84° in 2θ corresponding to the (112), (200), (103), (211) (004), (220), and (105) planes.

Additionally, the crystallite size for the nanomaterials was calculated using Scherrer’s equation shown in Equation (3) using the three most intense peaks from each diffractogram.
(3)$$D=\frac{k\lambda }{\beta \cos\theta }$$
where *D*, *k*, *λ*, *β*, and *θ* are the crystallite size, shape factor (0.9), X-ray wavelength, peak’s FWHM, and Bragg angle, respectively. The calculated crystallite size, lattice parameters, and χ^2^ can be seen in Table 2. Based on the χ^2^, the Le Bail fittings are in good agreement with literature data [14,15,19,26].

Figure 2 and Figure 3 show the SEM and TEM images for the hybrid ferrite, ZnO, and Mn_3_O_4_ NMs, respectively. Both the SEM and TEM images showed similar particle sizes and morphologies for the NMs, although the hybrid ferrite materials were more varied in morphology. For example, Fe_3_O_4_ and MnFe_2_O_4_ exhibited clustered globular/spherical particles ( Figure 2A,B and Figure 3A,B), while the Zn_0.5_Mn_0.5_Fe_2_O_4_ ( Figure 2D and Figure 3D), Mn_3_O_4_ ( Figure 2E and Figure 3E), and ZnO ( Figure 2F and Figure 3F) were comprised of aggregated platelet particles. The ZnFe_2_O_4_ ( Figure 2C and Figure 3C) nanomaterial consisted of pyramidal and irregular particles. The Fe_3_O_4_, MnFe_2_O_4_, ZnFe_2_O_4_, Zn_0.5_Mn_0.5_Fe_2_O_4_, Mn_3_O_4_, and ZnO had approximate particle sizes of 30, 25, 30, 40, 50, and 35 nm, respectively. There were some variations in particle sizes, which can be attributed to the lack of use of surfactants and capping agents during synthesis.

The hydrodynamic size and zeta-potential was determined for all nanomaterials and are shown in Table 3. The Fe_3_O_4_ and Mn_3_O_4_ exhibited a lower magnitude zeta-potential at +5.1 and −7.5 mV, respectively. Moreover, these two nanomaterials were highly aggregated in solution, having hydrodynamic sizes of 554.7 and 651 nm for Fe_3_O_4_ and Mn_3_O_4_, respectively. The other nanomaterials exhibited greater zeta-potentials and smaller hydrodynamic sizes, indicating more stability and dispersion in solution.

Fe, Mn, and Zn are considered important micronutrients required for proper plant growth and development. Therefore, magnetite (Fe_3_O_4_), hybrid ferrites (MnFe_2_O_4_, ZnFe_2_O_4_, and Zn_0.5_Mn_0.5_Fe_2_O_4_), manganese oxide (Mn_3_O_4_), and zinc oxide (ZnO) were designed and applied to Candyland Red tomatoes to evaluate their effects on the plant growth, fruit production, and fruit quality.

### 3.2. Plant Production

There were no differences of statistical significance between the control and the treatments in growth endpoints (root length, root weight, stem length, stem weight, and leaf weight) and fruit production (*p* > 0.5) (Table 4); however, there were minor increases in root weight and leaf weight when plants were exposed to several of the treatments. For example, MnFe_2_O_4_, ZnO, and all their ionic counterparts increased the root weights and leaf weights by approximately 26.5% and 13.0%, respectively. Several studies involving the exposure of metal oxide nanomaterials on tomato plants have shown similar nonsignificant effects. Cantu et al. found that there was no net effect on the agronomical parameters when tomato plants were exposed to CuO NPs and grown to full maturity (120 days) [16]. Adisa et al. and Barrios et al. also showed that soil or foliar application of CeO_2_ NPs on tomato did not alter plant growth at concentrations below 250 mg/kg and 250 mg/L when grown for 126 and 210 days, respectively [11,12]. Conversely, Raliya et al. found that exposure via foliar application of TiO_2_ on tomato plants cherry super sweet 100 variety decreased root length while ZnO increased the root length when exposed to concentrations up to 250 mg/L; higher concentrations of ZnO (500–1000 mg/L) inhibited root growth [10]. Moreover, Velasco et al. found that exposing tomato to either hexagonal ZnO or maltodextrin-coated ZnO at 1500 ppm via both a soil drench method and foliar application increased both the plant height and leaf weight [27]. In a study with seedlings, Lopez-Moreno et al. exposed tomato seedlings to CoFe_2_O_4_ NMs at various concentrations (62.5–1000 mg/L) and found a concentration-dependent increase in root length and decrease in stem length [28]. The variation in results may be attributed to species differences or to differences in experimental design and dosing regimen. For instance, Velasco et al. observed improvement in plant physiological responses at higher concentrations of ZnO [27]. Moreover, it is likely that as the duration of the tomato plant growth is extended, low concentrations of NMs will exert less of an effect on the physiological parameters; however, extensive studies are required to confirm this hypothesis.

### 3.3. Micronutrient Analysis

Micronutrients are essential elements required in small concentrations for proper growth and development. They are used in several processes such as photosynthesis, cellular respiration, and various defense responses [29]. For instance, iron is present in several organelles including the chloroplasts, mitochondria, and vacuoles [30,31]. It participates in chlorophyll biosynthesis, photosynthesis, cellular respiration, lipid metabolism, and tricarboxylic acid cycle (TCA) [32,33]. Similarly, manganese is involved in several metabolic processes, stress tolerance, and in the photosynthetic process [34]. It is involved in the activation of several enzymes and the production of various metabolites including Mn-SOD, Mn-CAT, chlorophyll, flavonoids, and lignin [34,35,36]. Zinc participates in the carbohydrate, protein, and pollen production, the activation of photosynthetic metabolism, and oxidative stress protection [33,37]. Cu plays a role in ROS mitigation, photosynthesis, phenol metabolism, and protein synthesis [33,38]. As shown in Figure 4, the majority of treatment effects in micronutrient concentrations were observed in the leaves. For instance, ZnFe_2_O_4_ increased the Cu, Fe, and Zn concentrations in the leaf tissues by 71.2%, 68.6%, and 138%, respectively. Similarly, Zn_0.5_Mn_0.5_Fe_2_O_4_ caused an increase in Fe and Zn by 81.6% and 90.9% in the leaves. A similar effect was observed for the Mn_3_O_4_ treatment, where Cu and Fe was increased, while Fe_3_O_4_ increased only the Fe content. Interestingly, ZnFe_2_O_4_ also increased the Cu concentration in the fruit. Previous studies involving metal oxide NMs have shown a wide range of impacts. For instance, Lopez-Vargas et al. observed that foliar exposure to Cu NPs at various concentrations did not cause an increase in Cu content in the fruits [39]. Similarly, Adisa et al. found that foliar application of CeO_2_ NMs did not cause the accumulation or translocation of micronutrients across the various tissues in tomato plants [12]. Conversely, Raliya et al. found that there was an accumulation of Zn in the leaves of tomato plants exposed to ZnO via foliar application [10]. The accumulation of the Fe, Mn, and Zn in the leaves may be due to the nanomaterials’ uptake by the plants via various leaf apertures (stomata, trichomes, and hydathodes) when applied foliarly [40]. The regulation of micronutrients occurs through different transport pathways which migrate the nutrients to the different areas needed. Ghasemi et al. found that certain nutrients may cause sensitivity towards other microelements and cause accumulation in various tissues [41]. This may explain the accumulation of Cu in the leaves and fruit when plants were exposed to ZnFe_2_O_4_.

### 3.4. Leaves Chlorophyll Production

The chlorophyll content in the tomato plants was measured one week after each application and at harvest (Figure 5). There was an increase in chlorophyll one week after the first treatment of the iron NMs (Fe_3_O_4_, MnFe_2_O_4_, ZnFe_2_O_4_, and Zn_0.5_Mn_0.5_Fe_2_O_4_); however, Fe^3+^, ZnO, Mn_3_O_4_, Zn^2+^, and Mn^2+^ were statistically equivalent to the controls. Moreover, after the second treatment and at harvest, there were no significant differences between the treatments and the controls. Similar findings were reported by Cantu et al., Adisa et al., and Barrios et al., where the nanotreatments had no effect on the chlorophyll content of Candyland red tomatoes (CuO and CuO-CA), Heirloom tomatoes (CeO_2_), and Roma tomatoes (CeO_2_ and CeO_2_-CA), at the end of the life cycle [11,12,16]. However, when the tomato was exposed to CuO and Al_2_O_3_ NMs in the soil for 40 days, increased chlorophyll content was reported [42]. Moreover, Yue et al. noted that there was an increase in photosynthetic rate and in both chlorophyll a and chlorophyll b in tomato seedlings and 40-day-old tomato plantlets, respectively [8]. The variability between the current study and previous studies may be attributed to a complex interplay of NM type, concentration, plant species, and growth cycle duration. Interestingly, in longer growth cycles, there seems to be less of an effect from NM exposure on tomato plant physiological parameters. However, a further comprehensive investigation at the molecular and genomic level is required to identify the plant–NM interactions.

### 3.5. Carbohydrate Quantification

Sugar, starch, and fiber are the main carbohydrates and energy sources within fruits and vegetables. As shown in Figure 6, the iron treatments had no net effect on the sugar content. However, the Mn_3_O_4_ and ZnO NMs, as well as their ionic counterparts, increased the sugar concentration by over 100% compared to controls. For starch, there were no statistical differences between the control and the various treatments. Previous studies have shown variable results when tomato plants are exposed to metal oxide NMs. For instance, when tomato plants were exposed to MnFe_2_O_4_ NMs, there was an increase in the sucrose and fructose precursor G6P, indicating that the NMs caused an increase in sugar content [8]. Barrios et al. and Adisa et al. found antagonist effects on sugar content from tomato fruits exposed to CeO_2_ NMs and CeO_2_-CA-functionalized NMs [43,44]. Moreover, soil application of CuO-CA-functionalized NMs decreased starch concentration in tomato fruit [16]. The observed increase in sugar content from exposures to Mn_3_O_4_ and ZnO may be a response to stress. Similar results were reported by Zhao et al., in which the upregulation of sucrose production in cucumbers treated with CeO_2_ NMs may occur as a sign of stress [45]. Moreover, Bolouri et al. found that sugars such as sucrose were important to stress-related responses [46]. Therefore, more comprehensive studies are required to develop an understanding of how these nanomaterials are altering the biochemical responses.

### 3.6. Bioactive Compounds Quantification

In general, the lycopene and β-carotene increased upon a 15-day storage at room temperature as shown in Figure 7, which can be attributed to a further ripening of the tomato fruits. However, the lycopene content in the 0-day-stored fruits from MnFe_2_O_4_-, ZnFe_2_O_4_-, and Zn_0.5_Mn_0.5_Fe_2_O_4_-treated plants was reduced relative to controls. However, by 15 days, the differences as a function of treatment had disappeared. There were trends for decreased content in the various iron treatments, although the differences were not statistically significant. The β-carotene increased upon storage time (Figure 7B) but again, at day 0 of storage, Zn_0.5_Mn_0.5_Fe_2_O_4_, Mn_3_O_4_, Mn^2+^, ZnO, and Zn^2+^ treatment values were significantly lower than the controls. In this instance, the negative effects of Mn_3_O_4_, Mn^2+^, ZnO, and Zn^2+^ were still evident in the reduced production of β-carotene throughout storage. Several previous studies have shown opposite effects from metal-based NM exposure. For instance, copper nanomaterials caused an increase in fruit lycopene concentrations in several reports [39,47,48]. Similar behavior was observed from foliar exposure of TiO_2_ and ZnO at various concentrations of 0–100 mg/kg and from both foliar and soil exposure of CeO_2_ NMs [10,43,44]. The primary carotenoid in tomato fruits is lycopene; this important biomolecule acts as an antioxidant and aids in the elimination of ROS [49,50]. Through the carotenoid biosynthesis pathway, lycopene may be converted to β-carotene or δ-carotene via lycopene beta-cyclase (β-LCY) or lycopene epsilon cyclase (ε-LCY), respectively [51]. Once converted into β-carotene, it may be further converted into xanthophylls such as zeaxanthin, antheraxanthin, violaxanthin, neoxanthin, and abscisic acid [52]. Moreover, β-carotene is a precursor of vitamin A, therefore, a decrease in β-carotene can have significant negative effects [53]. Vitamin A deficiency (VAD) is a health concern, particularly in developing countries [51]. The results from this study indicate that MnFe_2_O_4_, ZnFe_2_O_4_, and Zn_0.5_Mn_0.5_Fe_2_O_4_ inhibited the production of lycopene at the precursor stage, whereas, Zn_0.5_Mn_0.5_Fe_2_O_4_, Mn_3_O_4_, MnCl_2_, ZnO, and Zn(NO_3_)_2_ inhibited the production of β-carotene.

As shown in Figure 8, between 0 and 15 storage days, there was a decrease in total phenolic compounds. Comparing the treatments and the control at zero-day-stored, no differences of statistical significance were evident. Upon storage, the total phenolic compounds in the control decreased by 41%; however, this decrease was mitigated in the ZnFe_2_O_4_, Zn_0.5_Mn_0.5_Fe_2_O_4_, ZnO, FeCl_3_, MnCl_2_, and Zn(NO_3_)_2_ treatments and increased the total phenolic compounds by over 50% compared to the 15-day-stored control. Similar effects were observed in other studies with plants exposed to different metal nanomaterials. For instance, when a mixture of 4000 mg/L Zn and 2000 mg/L Cu NPs were introduced foliarly to basil plants, the total phenolic compounds were increased by approximately 25% [54]. Similarly, Hernandez-Fuentes et al. found an increase in total phenolic compounds in tomato plants exposed to Cu NPs with prolonged storage time [48]. Lopez-Vargas et al. also observed a slight increase in total phenolic compounds in tomato when treated with 250 mg/L Cu NPs [39]. However, Akanbi-Gada et al. found an antagonist and concentration (300–1000 mg/kg)-dependent effect between total phenolic compounds and ZnO nanomaterials in exposed fruit [9]. The flavonoid content is shown in Figure 9. The flavonoids of the control at zero-day-stored were approximately 8.7 mg/kg; the treatments were statistically equivalent to this value. That value had decreased by 80% after 15 days of storage across the control and all treatments. Previous studies have shown variable results on the flavonoid content in tomato fruits exposed to different metal-based nanomaterials. For instance, Yue et al. reported that exposing tomato plants to 10 mg/L of MnFe_2_O_4_ for four days consecutively enhanced the production of various metabolites, including rutin and quercetin [8]. Wang et al. exposed tomato plants to Fe_7_(PO_4_)_6_ nanomaterials at 5 and 50 mg/kg via soil application and also observed an increase in quercetin, rutin, and naringenin content at both doses [55]. Similarly, Cu-based nanomaterials have been shown to increase the flavonoid content in tomato fruits [39,48]. However, tomato plant exposure to ZnO nanomaterials via soil application resulted in a dose-dependent decrease in flavonoid content [9]. Similarly, Noori et al. also found a decrease in flavonoids from tomato fruits exposed to Ag NPs [56]. Phenolic compounds act as nonenzymatic antioxidants and are precursors to various secondary metabolites via the shikimic acid pathway [57]. The main phenolic compounds in tomato fruits are hydroxycinnamic acids, caffeic acid, chlorogenic acid, and flavonoids, including rutin, quercetin, and naringenin [58]. Both the total phenolic compounds and total flavonoids play an important role in plant stress tolerance [59]. In fruits, they aid in the diminution of free radicals and an increased presence of these phytonutrients may be attributed as a response to the formation of ROS from the nanomaterials. Overall, this observed increase in total phenolic compounds is beneficial for fruit quality; however, mass spectroscopy studies are required in the future to assess which metabolic species are being altered.

## 4. Conclusions

Overall, this study revealed that nanoscale treatments had generally nonsignificant effects on tomato physiological parameters and fruit production. However, there were some notable impacts on nutritional quality of the tomato fruits. For instance, the sugar content was enhanced in fruits of plants treated with Mn_3_O_4_ and ZnO NMs. The total phenolic compounds were enhanced by ZnFe_2_O_4_, Zn_0.5_Mn_0.5_Fe_2_O_4_, ZnO, and the iron and zinc ionic counterparts. Moreover, MnFe_2_O_4_, ZnFe_2_O_4_, and Zn_0.5_Mn_0.5_Fe_2_O_4_ initially slowed lycopene production, although after 15 days of storage at room temperature, this effect had disappeared. Lastly, β-carotene concentration decreased in fruits of plants treated with Mn_3_O_4_, ZnO, Mn^2+^, and Zn^2+^ when stored for both 0 and 15 days at room temperature. This indicates that there was an inhibition in the carotenoid biosynthesis pathway. This study demonstrates that NMs may induce both beneficial and detrimental effects in nutritional quality of tomato fruit. This highlights that caution is needed in such approaches and that further comprehensive studies are required to investigate and understand the underlying molecular basis of these impacts.

## Figures and Tables

**Figure 1 nanomaterials-12-02349-f001:**
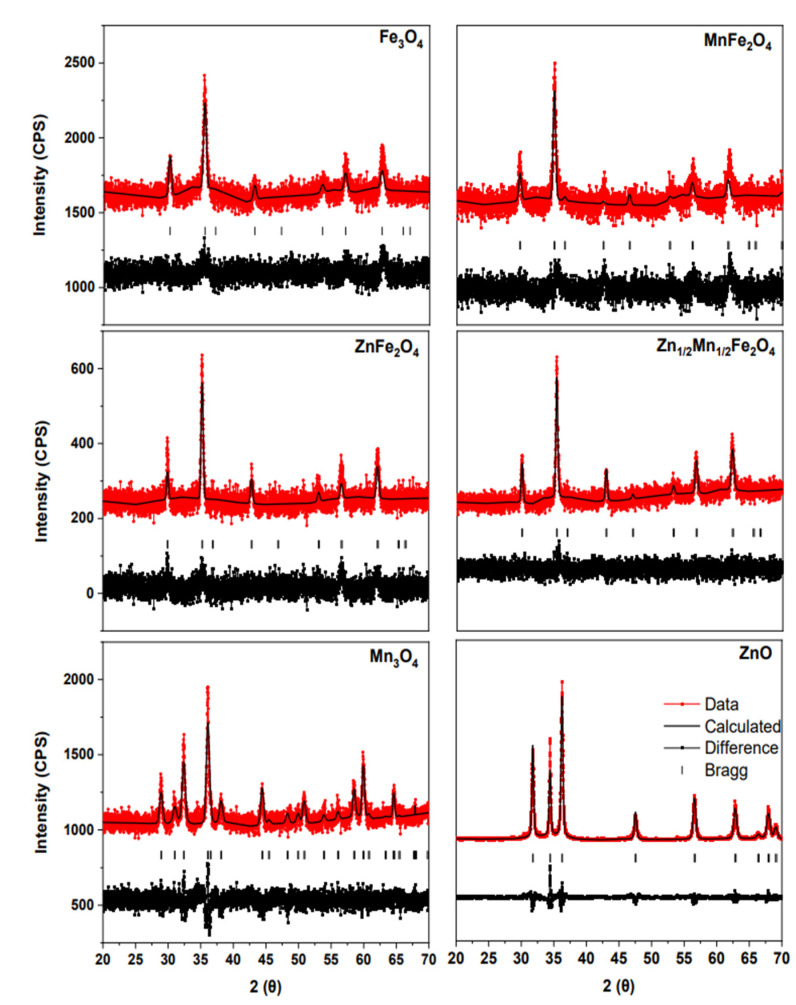
Fitted X-ray diffraction pattern for Fe_3_O_4_, MnFe_2_O_4_, ZnFe_2_O_4_, Zn_0.5_Mn_0.5_Fe_2_O_4_, Mn_3_O_4_ and ZnO.

**Figure 2 nanomaterials-12-02349-f002:**
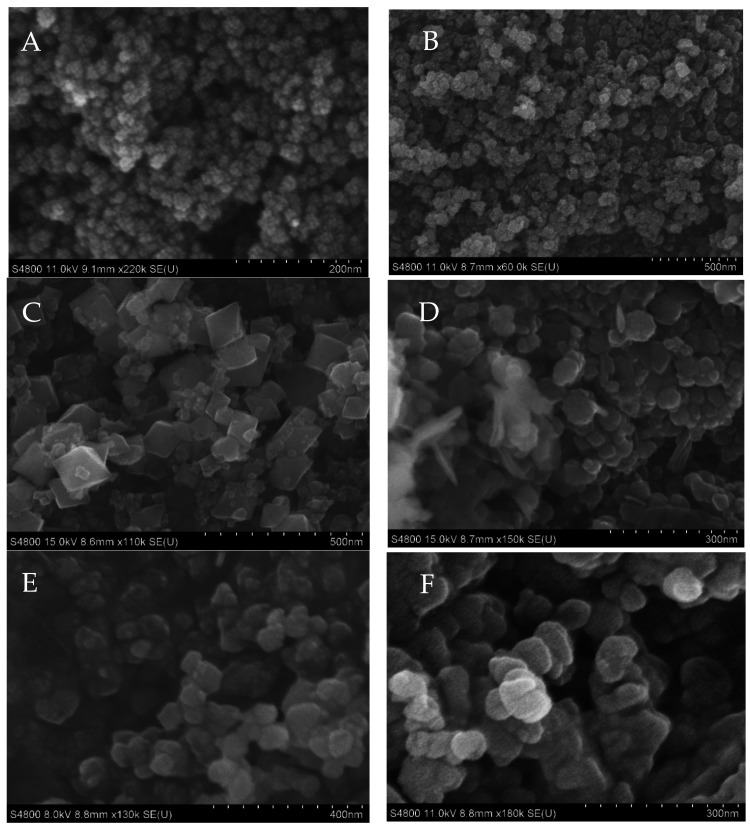
SEM Images of (**A**) Fe_3_O_4_, (**B**) MnFe_2_O_4_, (**C**) ZnFe_2_O_4_, (**D**) Zn_0.5_Mn_0.5_Fe_2_O_4_, (**E**) Mn_3_O_4_, and (**F**) ZnO.

**Figure 3 nanomaterials-12-02349-f003:**
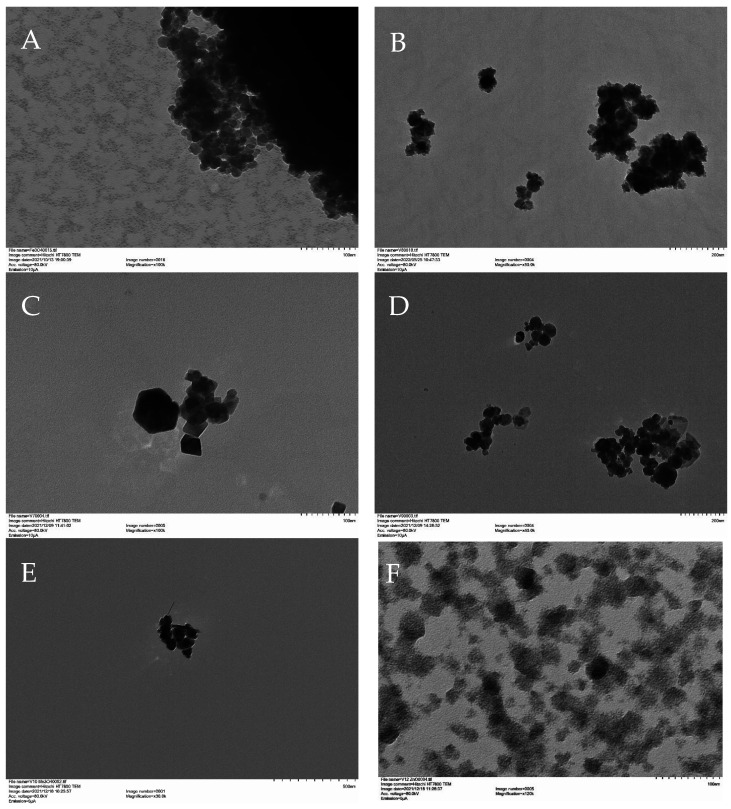
TEM Images of (**A**) Fe_3_O_4_, (**B**) MnFe_2_O_4_, (**C**) ZnFe_2_O_4_, (**D**) Zn_0.5_Mn_0.5_Fe_2_O_4_, (**E**) Mn_3_O_4_, and (**F**) ZnO.

**Figure 4 nanomaterials-12-02349-f004:**
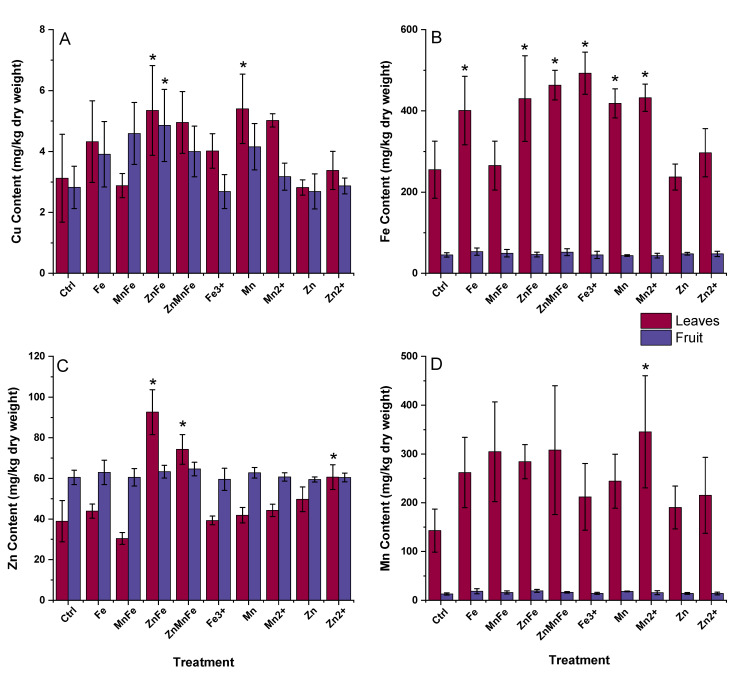
Effects of the Fe_3_O_4_ (Fe), MnFe_2_O_4_ (MnFe), ZnFe_2_O_4_ (ZnFe), Zn_0.5_Mn_0.5_Fe_2_O_4_ (ZnMnFe), Mn_3_O_4_ (Mn), ZnO (Zn), or their ionic counterparts on the (**A**) Cu content, (**B**) Fe content, (**C**) Zn content, and (**D**) Mn content in the leaves and fruit; n = 4; * denotes statistical difference between control and treatments.

**Figure 5 nanomaterials-12-02349-f005:**
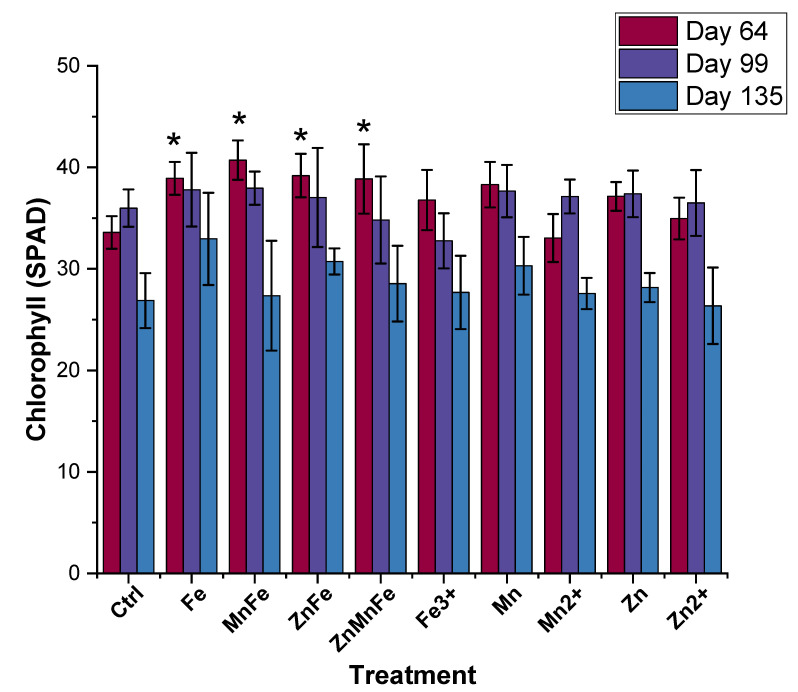
Chlorophyll content of tomato plant leaves after exposure to either Fe_3_O_4_ (Fe), MnFe_2_O_4_ (MnFe), ZnFe_2_O_4_ (ZnFe), Zn_0.5_Mn_0.5_Fe_2_O_4_ (ZnMnFe), Mn_3_O_4_ (Mn), ZnO (Zn), or their ionic counterparts at 64, 99, and 135 days of growth; n = 4; * denotes statistical difference between control and treatments.

**Figure 6 nanomaterials-12-02349-f006:**
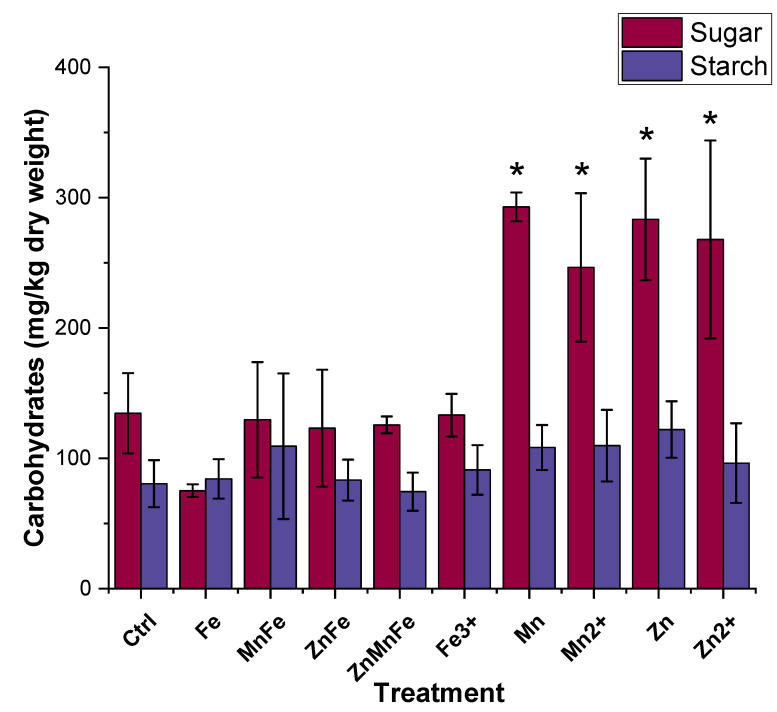
Tomato fruit carbohydrate profile after plant exposure to either Fe_3_O_4_ (Fe), MnFe_2_O_4_ (MnFe), ZnFe_2_O_4_ (ZnFe), Zn_0.5_Mn_0.5_Fe_2_O_4_ (ZnMnFe), Mn_3_O_4_ (Mn), ZnO (Zn), or their ionic counterparts; n = 4; * denotes statistical difference between control and treatments.

**Figure 7 nanomaterials-12-02349-f007:**
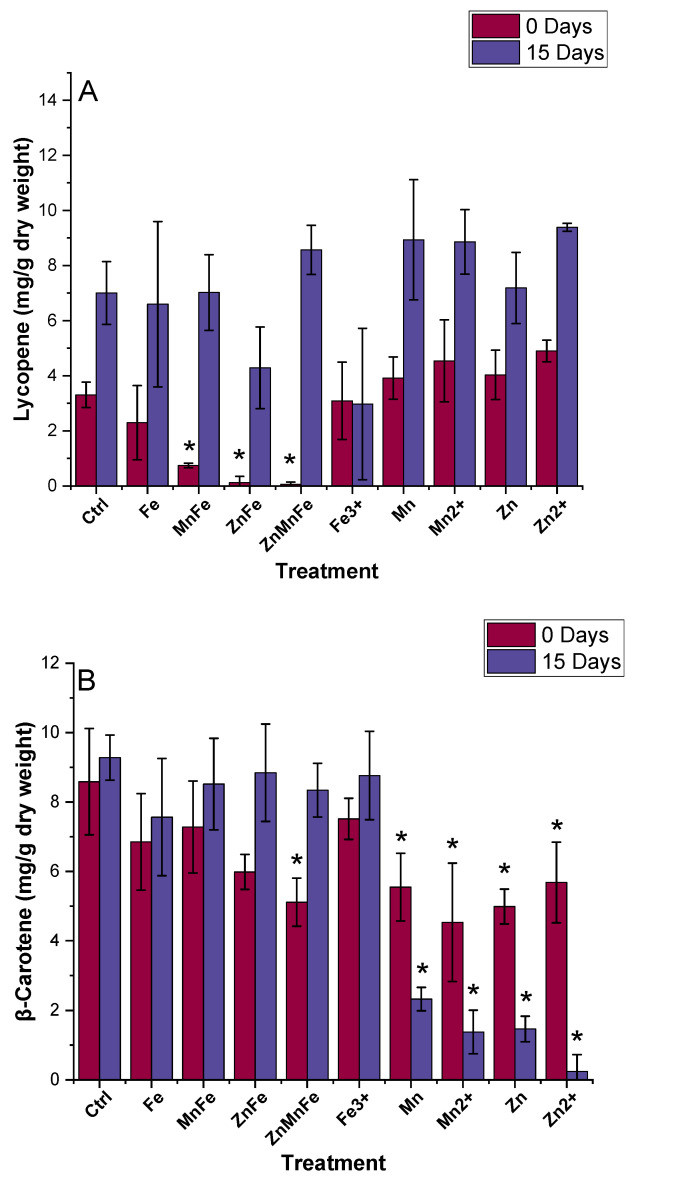
Effects of Fe_3_O_4_ (Fe), MnFe_2_O_4_ (MnFe), ZnFe_2_O_4_ (ZnFe), Zn_0.5_Mn_0.5_Fe_2_O_4_ (ZnMnFe), Mn_3_O_4_ (Mn), ZnO (Zn), and their ionic counterparts on the (**A**) lycopene and (**B**) β-carotene concentrations at 0 and 15 stored days; n = 4; * denotes statistical difference between control and treatments.

**Figure 8 nanomaterials-12-02349-f008:**
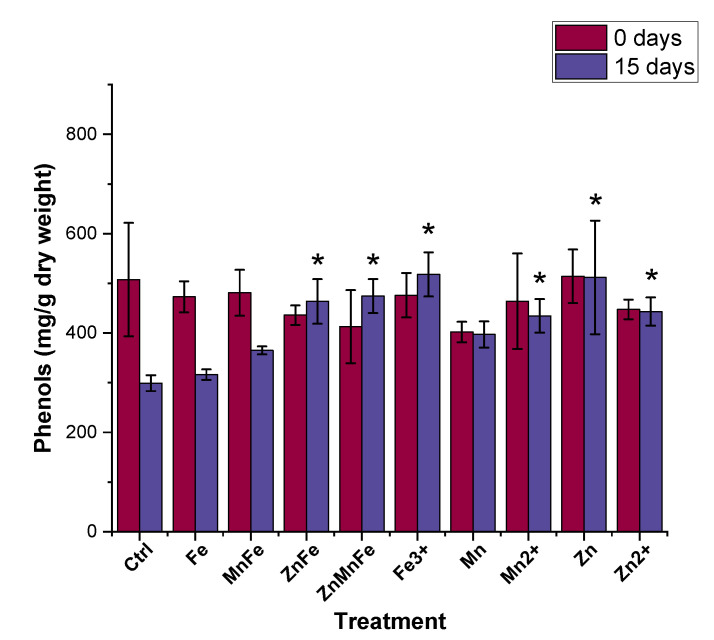
Total phenolic compounds content of tomato fruits stored at room temperature for 0 and 15 days after treatment with either Fe_3_O_4_ (Fe), MnFe_2_O_4_ (MnFe), ZnFe_2_O_4_ (ZnFe), Zn_0.5_Mn_0.5_Fe_2_O_4_ (ZnMnFe), Mn_3_O_4_ (Mn), ZnO (Zn), or their ionic counterparts; n = 4; * denotes statistical difference between control and treatments.

**Figure 9 nanomaterials-12-02349-f009:**
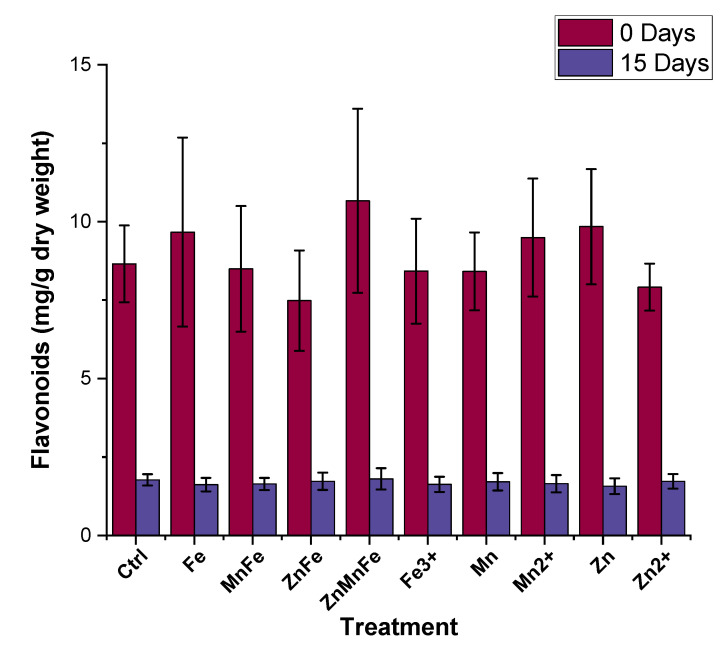
Flavonoid content of tomato fruits stored at room temperature for 0 and 15 days after treatment with either Fe_3_O_4_ (Fe), MnFe_2_O_4_ (MnFe), ZnFe_2_O_4_ (ZnFe), Zn_0.5_Mn_0.5_Fe_2_O_4_ (ZnMnFe), Mn_3_O_4_ (Mn), ZnO (Zn), or their ionic counterparts; n = 4.

**Table 1 nanomaterials-12-02349-t001:** Nanomaterial precursor concentrations and source.

Nanomaterial	Concentration and Precursor	Precursor Source
Fe_3_O_4_	10 mM Fe^3+^ and 20 mM Fe^2+^	FeCl_3_·6H_2_O and FeCl_2_·4H_2_O
MnFe_2_O_4_	10 mM Mn^2+^ and 20 mM Fe^3+^	MnCl_2_·4H_2_O and FeCl_3_·6H_2_O
ZnFe_2_O_4_	10 mM Zn^2+^ and 20 mM Fe^3+^	Zn(NO_3_)_2_∙6H_2_O and FeCl_3_∙6H_2_O
Zn_0.5_Mn_0.5_Fe_2_O_4_	5 mM Zn^2+^, 5 mM Mn^2+^, and 20 mM Fe^3+^	Zn(NO_3_)_2_∙6H_2_O, MnCl_2_∙4H_2_O, and FeCl_3_∙6H_2_O
Mn_3_O_4_	30 mM Mn^2+^	MnCl_2_·4H_2_O
ZnO	60 mM Zn^2+^	Zn(NO_3_)_2_∙6H_2_O

**Table 2 nanomaterials-12-02349-t002:** Lattice parameters for various nanomaterials and calculated crystallite size.

**Material**	**Space Group**	**a**	**b**	**c**	**Lattice Angles**	**χ2**	**Crystallite Size (nm)**
**Fe_3_O_4_**	FD3M	8.3969	8.3969	8.3969	α = β = γ = 90	1.22	15.30 ± 0.85
**MnFe_2_O_4_**	FD3M	8.5115	8.5115	8.5115	α = β = γ = 90	1.24	17.70 ± 3.21
**ZnFe_2_O_4_**	FD3M	8.4479	8.4479	8.4479	α = β = γ = 90	1.37	22.68 ± 5.52
**Zn_0.5_Mn_0.5_Fe_2_O_4_**	FD3M	8.4296	8.4296	8.4296	α = β = γ = 90	1.07	24.64 ± 1.28
**Mn_3_O_4_**	I4_1_/AMD	5.7674	5.7674	9.4420	α = β = γ = 90	1.5	23.88 ± 1.33
**ZnO**	P6_3_MC	3.2524	3.2524	3.2524	α = β = 90, γ = 120	1.87	28.82 ± 6.72

**Table 3 nanomaterials-12-02349-t003:** Hydrodynamic size and zeta-potential for all nanomaterials.

Material	Hydrodynamic Size (nm)	Zeta-Potential (mV)	pH
Fe_3_O_4_	554.7 ± 17.28	+5.1 ± 0.2	6.21
MnFe_2_O_4_	433.5 ± 4.16	+23.9 ± 0.7	4.83
ZnFe_2_O_4_	160.3 ± 3.79	−29.3 ± 1.0	9.73
Zn_0.5_Mn_0.5_Fe_2_O_4_	230.9 ± 2.34	−38.2 ± 3.8	8.75
Mn_3_O_4_	651.3 ± 14.02	−7.5 ± 0.8	6.47
ZnO	358.0 ± 3.65	−16.7 ± 1.6	9.69

**Table 4 nanomaterials-12-02349-t004:** Tomato plant physiological parameters and fruit production after exposure to either Fe_3_O_4_ (Fe), MnFe_2_O_4_ (MnFe), ZnFe_2_O_4_ (ZnFe), Zn_0.5_Mn_0.5_Fe_2_O_4_ (ZnMnFe), Mn_3_O_4_ (Mn), ZnO (Zn), or their ionic counterparts; n = 4.

Treatment	Root Length (cm)	Root Weight (g)	Stem Length (cm)	Leaf Weight (g)	Total Fruits	Overall Fruit Weight (g)
Ctrl	56.0 ± 3.0	75.0 ± 31.0	241.2 ± 26.7	172.5 ± 38.7	25.0 ± 11.2	42.4 ± 17.1
Fe	60.6 ± 13.4	76.3 ± 41.1	230.3 ± 32.4	180.8 ± 39.4	26.5 ± 15.6	39.5 ± 22.3
MnFe	59.0 ± 10.4	90.5 ± 17.9	216.25 ± 21.8	197.5 ± 32.4	26.7 ± 12.3	36.2 ± 22.1
ZnFe	57.9 ± 4.2	46.8 ± 20.6	258.0 ± 23.4	171.5 ± 35.5	21.8 ± 9.0	31.6 ± 11.5
ZnMnFe	68.0 ± 10.4	76.6 ± 16.1	256.0 ± 9.0	183.0 ± 15.8	30.8 ± 9.03	44.1 ± 3.3
Fe^3+^	61.9 ± 9.2	96.8 ± 24.6	213.8 ± 26.0	200.0 ± 12.4	39.8 ± 17.7	52.5 ± 21.8
Mn	59.0 ± 8.4	78.2 ± 12.7	262.5 ± 19.2	178.8 ± 26.4	22.3 ± 9.7	28.4 ± 13.3
Mn^2+^	54.9 ± 14.8	93.0 ± 11.4	227.3 ± 5.4	164.8 ± 30.8	29.3 ± 5.0	41.9 ± 6.0
Zn	60.6 ± 10.6	96.8 ± 10.0	249.8 ± 17.2	196.8 ± 34.8	43.8 ± 7.0	64.8 ± 12.3
Zn^2+^	59.5 ± 3.3	97.4 ± 9.9	229.8 ± 27.0	192.5 ± 18.9	41.8 ± 12.7	55.3 ± 20.4
*p*-value	0.824	0.108	0.06	0.733	0.065	0.106

No statistical difference between treatments and control throughout different plant tissues.

## Data Availability

Data is contained within the article. The data presented in this study are available in the article.

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
