# Peer review of "Tomato Fruit Nutritional Quality Is Altered by the Foliar Application of Various Metal Oxide Nanomaterials"

_nanomaterials, 2022, doi:10.3390/nano12142349_

Round 1

Reviewer 1 Report

Cantu et al. demonstrate the possitive and negative effects of the use of a wide range of nanomaterials on tomato fruit quality over a period of 15 days. In the manuscript the authors report the fabrication and characterization of metal-based nanomaterials and the impact of the latter on the nutritional quality of the tomato fruit. Overall, the paper reach the quality to be accepted and only some minor issues need to be addressed.

I would advice the authors to introduce a short descriptio in the result section 3.1 about what is the motivation to asses these nanomaterials on plant development. Although it is clearly explain in the introduction it could help the reader to follow the full story. Regarding the effect of nanomaterials on plant growth I would appreciate to compare or go into deeper comparison of the numbers and quantifications they perform.

The study is performed in time frame of 15 days of storage, mostly all the effects dropped after 15 days. Can the authors predict an optimal storage time? i.e. if in a week nutrional improvement might be optimum?

The authors asses nanomaterials based on Fe, Zn and Mn. Would be interesting to asses nanomaterials based on Calcium , Sodium or Potassium?

Reviewer 2 Report

This is a very-well designed and written report.

I have no pertinent suggestions for the authors.

Author Response

Thank you for taking the time to review our manuscript!

Reviewer 3 Report

The manuscript “Tomato Fruit Nutritional Quality is Altered by the Foliar Application of Various Metal Oxide Nanomaterials” deals with plant priming with Mn-, Zn- and Fe-based NMs and the effect on fruit quality. This is an interesting study, clearly written, however, the discussion is a little bit superficial, lacking a mechanistic hypothesis. It could be improved.  

Abstract

Authors should correct the nomenclature of the materials used.

Introduction

L 59: b-carotene should be β-carotene 

Material and Methods: 

L113: remove 4

How many pots/plants were used per treatment? What volume was used in each plant in each application? This information must be included.

Why did the authors apply the suspensions after 43 and 78 d? 

Results

L253: “250 mg/L higher concentrations inhibited root growth” – something is missing

L273, 275: maybe the authors can replace "aids" for another word such as "participates"

L294: A “by” is missing

L331, 332: The three main carbohydrates and energy sources within fruits and vegetables are sugar, starch, and fiber.

L344-346: rephrase 
